# Necessity of Individualized Approach for Gastric Subepithelial Tumor Considering Pathologic Discrepancy and Surgical Difficulty Depending on the Gastric Location

**DOI:** 10.3390/jcm11164733

**Published:** 2022-08-13

**Authors:** Sung Gon Kim, Bang Wool Eom, Hongman Yoon, Myeong-Cheorl Kook, Young-Woo Kim, Keun Won Ryu

**Affiliations:** 1Center for Gastric Cancer, National Cancer Center, Goyang-si 10408, Korea; 2Department of Surgery, Konyang University Hospital, Daejeon 35365, Korea

**Keywords:** gastric subepithelial tumor, gastrointestinal stromal tumor, laparoscopic wedge resection

## Abstract

Background: Depending on the location of gastric subepithelial tumors (SETs), surgical access is difficult with a risk of postoperative complications. This study aimed to evaluate the clinicopathological characteristics of small-sized gastric SETs and their surgical outcomes depending on location and provide considering factors for their treatment plans. Methods: This single-center, retrospective study reviewed patients who underwent surgical resection for gastric SETs (size < 5 cm). SETs were divided into benign SETs and gastrointestinal stromal tumors (GISTs) for comparison. The clinicopathological characteristics of SETs in the cardia were compared to those in the other regions. Results: Overall, 191 patients with gastric SETs (135 GISTs, 70.7%; and 56 benign SETs, 29.3%) were included. In multivariate analysis, age > 65 years (odds ratio (OR), 3.183; 95% confidence interval (CI), 1.310–7.735; *p* = 0.011), and non-cardiac SETs (OR, 2.472; 95% CI, 1.110–5.507; *p* = 0.030) were associated with a significant risk of malignancy. Compared to SETs in other locations, cardiac SETs showed more complications (3 versus 0; *p* = 0.000), and open conversion rates (2 versus 0; *p* = 0.003). However, the proportion of GISTs of SETs in the cardia is not negligible (52.9%). Conclusions: Considering the malignancy risk of SETs, active surgical resection should be considered in old age and/or location in the non-cardiac area. However, in young patients, SETs located in the gastric cardia have a considerably benign nature and are associated with poor short-term surgical outcomes. An individualized surgical approach for asymptomatic small SETs according to the gastric location is warranted.

## 1. Introduction

Gastric subepithelial tumors (SETs) are rare lesions that account for less than 2% of all gastric tumors [1]. The majority of such lesions are small, asymptomatic, and accidentally found during routine upper endoscopy. With the increase in screening through upper endoscopy, gastric SETs are easily encountered [2,3]. The pathologic diagnosis of SETs is diverse, including gastrointestinal stromal tumors (GISTs), leiomyomas, schwannomas, heterotopic pancreas, and other benign or potentially malignant lesions. Among these, GISTs are the most common and potentially malignant lesions [4,5,6].

Because it is difficult to determine the pathologic diagnosis of SETs before surgery and the potential risk of malignancy, the standard treatment for gastric SETs is surgical resection with negative surgical margins [5,7,8]. In several retrospective studies, based on the location, pathological discrepancies in gastric SETs have been reported, and benign tumors such as leiomyomas are the most common neoplasms in the cardia [9,10,11]. After surgical resection, numerous small SETs in the cardia are diagnosed as benign tumors. The surgical resection of such small benign SETs is not mandatory, considering the postoperative risk.

With advancements in technology, laparoscopic wedge resection has been recognized as the optimal choice for the treatment of gastric SETs [1,12]. However, the resection of SETs located in the cardia is extremely difficult and may lead to the risk of several complications, such as leakage, stenosis, and reflux [13,14,15,16]. Sometimes, major gastric resections, such as proximal gastrectomy or total gastrectomy, are inevitable. Hence, there is still controversy regarding the diagnosis and treatment plan for SETs in the cardia [17,18]. Therefore, further consensus on the diagnosis and treatment of SETs in the cardia is required to reduce unnecessary invasive surgery.

In this study, we retrospectively reviewed the clinicopathological features of surgically resected small gastric SETs. Depending on the location, we analyzed the pathological discrepancy and surgical outcomes of SETs to identify the factors that determine the individualized surgical approach for gastric SETs.

## 2. Materials and Methods

### 2.1. Patients and Methods

A retrospective single-center study involved 191 consecutive patients who underwent surgical resection for gastric SETs (size < 5 cm) at the National Cancer Center, Korea, between January 2000 and April 2020. The data collected were patients’ demographics, operation records, and tumor characteristics, including tumor size, location, change in mucosa, growth pattern, and histopathological findings. Before surgical resection, all the patients were diagnosed with gastric SETs using esophagogastroduodenoscopy (EGD) and computed tomography (CT). When necessary, endoscopic ultrasound (EUS) was performed to estimate the depth of the lesion, whereas endoscopic biopsy was performed to assess the mucosal changes of the lesion. This study was approved by the Institutional Review Board of the National Cancer Center (approval number: NCC2021-0026).

### 2.2. Statistical Analysis

All statistical analyses were performed using R (version 2.12.1, R Foundation for Statistical Computing, Vienna, Austria). Categorical variables were analyzed using the chi-squared test, and continuous data were analyzed using Student’s t-test and were described as mean plus minus standard deviation (mean ± SD). Multivariate analysis was performed using logistic regression analysis to assess the predictive risk factors for GISTs. Statistical significance was considered as a *p*-value less than 0.05.

## 3. Results

### 3.1. Baseline Characteristics of Patients

A total of 191 patients with gastric SETs (less than 5 cm in diameter) who underwent surgical resection were included in this study (Table 1). The mean age was 57.3 ± 11.4 years (range 24–85 years); 83 (43.5%) patients were men, and 108 (56.5%) patients were women. The mean size of SETs was 2.8 ± 0.9 cm (range 0.5–4.9 cm), and 34 SETs (17.85%) were located in the cardia. Wedge resection (laparoscopic resection) was performed in 183 patients (95.8%) and gastrectomy (open resection) in 8 patients (4.2%). One in eight patients who underwent gastrectomy were converted during wedge resection, whereas two of the eight patients underwent open surgery during laparoscopic wedge resection. According to the National Institutes of Health classification, among all SETs, 135 patients (70.7%) were histologically diagnosed with GISTs, and 56 patients (29.3%) were diagnosed with benign SETs, including leiomyomas (18, 9.4%), schwannomas (15, 17.8%), heterotopic pancreas (14, 7.3%), and other benign SETs (9, 4.7%), such as inflammatory pseudo-tumor, fibrotic nodule, and lymphoid hyperplasia. Seventy-three patients underwent endoscopic biopsy or endoscopic ultrasound-guided fine-needle aspiration (EUS-FNA) preoperatively, and 13 patients (17.8%) were diagnosed with GISTs or suspicious findings of GIST (data not shown).

### 3.2. Comparison of Benign SET and GIST/Risk Factors for GISTs

While comparing benign SETs and GISTs, the mean age was significantly higher in the GIST group (51.2 ± 10.9 versus 59.8 ± 10.7; *p* = 0.000), and the proportion of location in the cardia was significantly higher in benign SETs (28.6% versus 13.3%; *p* = 0.004). There were no differences in tumor size, growth pattern, and mucosal changes (Table 2). In multivariate analysis, for risk factors of GIST, age > 65 years was a significant risk factor (odds ratio (OR) 3.183; 95% confidence interval (CI): 1.310–7.735; *p* = 0.011), and tumor location in non-cardia showed a significantly higher odds ratio than other locations (OR 2.472; 95% CI: 1.110–5.507; *p* = 0.030) (Table 3).

### 3.3. Comparison of Cardiac SETs and SETs in Other Locations

A total of 34 SETs were located at the cardia compared to SETs in other locations (Table 4). The mean age was significantly higher in non-cardiac SETs (52.6 ± 13.6 versus 58.3 ± 10.6; *p* = 0.026). Cardiac SETs appeared larger in size (3.1 ± 1.0 cm versus 2.8 ± 0.9 cm; *p* = 0.041) and showed a more endophytic growth pattern (73.5% versus 53.5%; *p* = 0.032) than non-cardiac SETs. The proportion of benign SETs was significantly higher in cardiac SETs (47.0% versus 25.5%; *p* = 0.012) than in the non-cardiac group. Eight patients (4.2%) underwent open surgery, and two of them underwent open conversion during laparoscopic resection due to difficulty in resection and failure in primary closure after wedge resection. There were more intraoperative complications in cardiac SETs (4 [11.8%] versus 6 [3.8%]; *p* = 0.069) compared to three complications postoperatively. In addition, the mean operation time (148.1 ± 70.2 min versus 84.9 ± 44.0 min; *p* = 0.000) and the mean hospital stay (6.4 ± 2.6 days versus 5.2 ± 1.9 days; *p* = 0.001) were significantly longer in cardiac SETs.

### 3.4. Complications

There were ten intraoperative complications and three postoperative complications. Intraoperative complications consisted of five bleeding events at the resection site (one in cardiac SETs, four in non-cardiac SETs), three gastric serosal injuries (one in cardiac SETs, two in non-cardiac SETs), and two perforations at the suture line after resection of cardiac SETs. Postoperative complications included cases of suspected micro-perforations at the stapled line, which were treated conservatively.

## 4. Discussion

In this retrospective study, we reviewed 191 gastric SETs and evaluated the clinicopathological differences of SETs in the cardia compared to those of SETs in the other locations. Our results reveal that, in the cardiac SETs, the proportion of benign tumors was significantly higher, with more perioperative complications, a longer operation time, and prolonged hospital stay postoperatively. In a multivariate analysis, an age over 65 years and the location of the non-cardiac area were significant risk factors for GISTs.

Gastric SETs exhibit various clinical courses, ranging from benign to malignant. Most SETs are small, asymptomatic, and clinically insignificant at the time of detection [4,19]. Because these lesions arise from the muscles of neural origin and are covered with normal gastric mucosa, it is difficult to histologically diagnose them before surgical resection. Preoperative CT scan, EGD, or EUS have limitations in determining whether these lesions are benign or potentially malignant [19]. Therefore, surgical resection is usually recommended for gastric SETs larger than 2 cm [5,20]. Numerous patients undergo surgery without a definite histological diagnosis; however, controversy persists over the diagnostic and treatment plans for gastric SETs.

SETs located in the cardia are extremely difficult to surgically access, and there are concerns regarding postoperative complications such as luminal leakage, stenosis, and gastroesophageal reflux [16,21,22]. However, in patients who have undergone surgery, a significant number of benign SETs (not requiring immediate surgery) are sometimes diagnosed postoperatively. Several studies have reported that cardiac SETs are difficult to surgically resect with adequate margins, and postoperative complications, such as gastroesophageal reflux, stenosis, or leakage, may occur [1,13,15]. Our results also show a higher complication rate in cardiac SETs. Furthermore, two patients underwent open conversion during laparoscopic surgery. One of them failed to suture the gastric lumen after laparoscopic wedge resection and was converted to open surgery. The gastric lumen was sutured ineffectively after conversion, and proximal gastrectomy with double-tract reconstruction was performed. Although only two patients underwent open conversion, the results indicate that the resection of SETs in the cardia may increase the risk of open conversion during minimally invasive surgery.

Lee et al. [9] reported histological characteristics of gastric SETs based on their location. The results reveal that, in the cardia, the characteristics of gastric SETs were significantly different from those at the other locations. Several studies reported that leiomyomas were the most common SET in the cardia. The results of this study show that benign SETs were more frequent in the cardia than in other locations (47.0% versus 25.5%; *p* = 0.012), and all the 16 benign SETs were histologically diagnosed as leiomyomas. However, nearly 50% of the cardiac SETs are still GISTs. Therefore, ignorance of malignancy at the cardia is alarming.

There are controversies regarding the diagnosis and treatment of gastric SETs [17,19]. Hence, most patients undergo surgery based on the size of the tumor or the presence of symptoms. Sometimes, surgeons find it difficult to perform immediate surgical procedures for SETs at difficult locations due to the risk of postoperative complications. Therefore, the literature recommends a tailored approach for SETs in the cardia [23,24], but there are concerns about the difficulty and complications of surgery.

Because benign SETs are the most common tumors in the cardia, there should be an accurate histological diagnosis before surgery. Several studies have reported the safety and feasibility of endoscopic ultrasound-guided fine-needle aspiration (EUS-FNA) and/or fine-needle biopsy (FNB) [25,26,27,28]. However, it is difficult to distinguish GISTs from benign tumors due to the insufficient amount of tissue obtained by EUS-FNA; therefore, it is not used as a routine diagnostic tool in gastric SETs [11,29,30]. As mentioned above, the cardia is a difficult and risky location to surgically access; such preoperative diagnostic tools might be helpful in establishing a treatment plan. If benign histological diagnosis can be confirmed before surgery, endoscopic surveillance without immediate surgery is a reasonable treatment plan for asymptomatic patients, and invasive surgery can be avoided.

This study has some limitations. First, it was a retrospective, single-institutional study with a limited sample size, and there might be selection bias. Second, since a number of patients have not undergone preoperative EUS-FNA and FNB in this study, further prospective studies are required to verify the efficacy of the preoperative pathological diagnosis via EUS-FNA and FNB.

In conclusion, old age and non-cardiac location were risk factors for GISTs in the multivariate analysis. SETs located in the cardia are difficult to surgically access, and surgery in this complicated location can lead to longer operation times, a prolonged hospital stay, and more complications. Therefore, a treatment plan for asymptomatic small-sized SETs should be individualized depending on the location of SETs. Because benign tumors are more frequent in the cardia than other locations, an accurate pathological diagnosis should be obtained preoperatively through further diagnostic tools such as EUS-FNA and FNB. Further studies for the better preoperative diagnosis of these diagnostic tools should be conducted in the future.

## Figures and Tables

**Table 1 jcm-11-04733-t001:** Baseline characteristics of patients.

Variable		Total
		*N* = 191
Age (years)	Mean ± SD	57.3 ± 11.4
Sex	Male	83 (43.5%)
	Female	108 (56.5%)
BMI (kg/m^2^)	Mean ± SD	24.8 ± 3.3
Size (cm)	Mean ± SD	2.8 ± 0.9
Location of SET	Cardia	34 (17.8%)
	Fundus	26 (13.6%)
	Body	109 (57.1%)
	Antrum	22 (11.5%)
Extent of resection	Wedge	183 (95.8%)
	Gastrectomy	8 (4.2%) ^a^
Surgical approach	Laparoscopy	183 (95.8%)
	Open	8 (4.2%) ^b^
Hospital stay (days)	Mean ± SD	5.4 ± 2.1
Adjuvant chemotherapy		19 (9.9%)
Pathology		
GIST		135 (70.7%)
NIH classification	Very low risk	41 (30.4%)
	Low risk	26 (19.3%)
	Intermediate risk	59 (43.7%)
	High risk	9 (3.8%)
Leiomyoma		18 (9.4%)
Schwannoma		15 (7.8%)
Heterotopic pancreas		14 (7.3%)
Others		9 (4.7%)

SD, standard deviation; SET, subepithelial tumor; GIST, gastrointestinal stromal tumor. ^a^ One patient underwent gastrectomy conversion during wedge resection. ^b^ Two patients underwent open conversion during laparoscopic approach.

**Table 2 jcm-11-04733-t002:** Comparison of benign SETs and GISTs.

Variable		Benign SET	GIST	*p*-Value
		*N* = 56	*N* = 135	
Age (years)	Mean ± SD	51.2 ± 10.9	59.8 ± 10.7	0.000
Sex	Male	24 (42.8)	59 (43.7)	0.914
	Female	32 (57.2)	76 (56.3)	
BMI (kg/m^2^)	Mean ± SD	24.7 ± 3.1	24.8 ± 3.3	0.783
Size (cm)	Mean ± SD	2.9 ± 1.0	2.8 ± 0.9	0.306
Growth pattern	Endophytic	36 (64.3)	73 (54.1)	0.194
	Exophytic	20 (35.7)	62 (45.9)	
Mucosal change	No	40 (71.4)	105 (77.8)	0.350
	Yes	16 (28.6)	30 (22.2)	
Location of SET	Cardia	16 (28.6)	18 (13.3)	0.004
	Fundus	1 (1.8)	25 (18.5)	
	Body	32 (57.1)	77 (57.0)	
	Antrum	7 (12.5)	15 (11.1)	
Extent of gastric resection	Wedge	51 (91.1)	132 (97.8)	0.087
	Gastrectomy	5 (8.9) ^a^	3 (2.2)	
Surgical approach	Laparoscopy	52 (92.9)	131 (97.0)	0.189
	Open	4 (7.1)	4 (3.0) ^b^	
Hospital stay (days)	Mean ± SD	6.0 + 2.1	5.2 + 2.0	0.018

SET, subepithelial tumor; GIST, gastrointestinal stromal tumor; SD, standard deviation. ^a^ One patient was converted to open proximal gastrectomy during laparoscopic wedge resection for SET at gastric cardia and was finally diagnosed with leiomyoma. ^b^ Two patients were converted to open surgery due to failure of laparoscopic wedge resection.

**Table 3 jcm-11-04733-t003:** Risk factors of GISTs in multivariate analysis.

Variable		Odds Ratio	95% CI	*p*-Value
Age	≥65	3.183	1.310–7.735	0.011
Sex	Male	1.045	0.537–2.034	0.896
Tumor location	Non-cardia	2.472	1.110–5.507	0.030
Growth pattern	Endophytic	0.441	0.390–1.507	0.441
Tumor size	>2 cm	1.270	0.541–2.981	0.583
Mucosal change	Yes	0.687	0.325–1.450	0.324

**Table 4 jcm-11-04733-t004:** Comparison of cardiac and non-cardiac SETs.

Variable		Cardia	Non-Cardia	*p*-Value
		*N* = 34	*N* = 157	
Age (years)	Mean ± SD	52.6 ± 13.6	58.3 ± 10.6	0.026
Sex	Male	15 (44.1)	68 (43.3)	0.932
	Female	19 (55.9)	89 (56.7)	
BMI (kg/m^2^)	Mean ± SD	24.9 ± 2.8	24.8 ± 3.4	0.919
Size (cm)	Mean ± SD	3.1 ± 1.0	2.8 ± 0.9	0.041
Mucosal change	No	25 (73.5)	120 (76.4)	0.615
	Yes	9 (26.5)	37 (23.6)	
Growth pattern	Endophytic	25 (73.5)	84 (53.5)	0.032
	Exophytic	9 (26.5)	73 (46.5)	
Diagnosis	Benign	16 (47.0)	40 (25.5)	0.012
	GIST	18 (52.9)	117 (74.5)	
Extent of gastric resection	Wedge resection	31 (91.2)	152 (96.8)	0.137
	Gastrectomy	3 (8.8)	5 (2.5)	
Surgical approach	Laparoscopy	30 (88.2)	153 (97.5)	0.015
	Open	4 (11.8)	4 (2.5)	
Open conversion		2 (5.9)	0 (0)	0.003
Operation time (minutes)	Mean ± SD	148.1 ± 70.2	84.9 ± 44.0	0.000
Complications				
Intraoperative		4 (11.8)	6 (3.8)	0.069
Postoperative		3 (8.8)	0 (0)	0.000
Hospital stay (days)	Mean ± SD	6.4 ± 2.6	5.2 ± 1.9	0.001

SD, standard deviation; BMI, body mass index; GIST, gastrointestinal stromal tumor.

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
