# Peer review of "Necessity of Individualized Approach for Gastric Subepithelial Tumor Considering Pathologic Discrepancy and Surgical Difficulty Depending on the Gastric Location"

_jcm, 2022, doi:10.3390/jcm11164733_

Round 1

Reviewer 1 Report

It is not specified if and how many of the patients underwent a preoperative histologic/citologic diagnosis, and how coherent this was with the post-operative pathological result. The ability of accurately diagnose the disease pre-operatively is the milestone of the treatment, therefore these information are fundamental.

Since "cardiac SETs appeared larger in size  and showed a more endophytic growth pattern" and were therefore more likely symptomatic than the others, surgical treatment must always be considered, and clinical observation not applicable. The stress in the conclusions should be on the necessity to find a better diagnostic tool, rather than hinting that surgical treatment could be postponed if not avoided. 

Author Response

Thank you for your kind comments.

We have revised the manuscript according to your comments. 

It is not specified if and how many of the patients underwent a preoperative histologic/citologic diagnosis, and how coherent this was with the post-operative pathological result. The ability of accurately diagnose the disease pre-operatively is the milestone of the treatment, therefore these information are fundamental.

--> It has been added to the results section. 

Since "cardiac SETs appeared larger in size  and showed a more endophytic growth pattern" and were therefore more likely symptomatic than the others, surgical treatment must always be considered, and clinical observation not applicable. The stress in the conclusions should be on the necessity to find a better diagnostic tool, rather than hinting that surgical treatment could be postponed if not avoided. 

--> As you pointed out, some of the conclusions have been modified.

Reviewer 2 Report

The research is significant as a result of detailing the small gastric SETs pathology depending on the epidemiological, clinical, endoscopic, morphological and surgical outcomes parameters. the univariate and multivariate analyses support an individualized treatment of patients adapted especially for cardiac localization.

The article addresses a topical issue as a result of the identification of some small lesions discovered by imaging methods, on which the therapeutic decision had to be adapted and correlated with the location through possible post-operative complications.

The study group contains a significant number of patients from the same medical center, and the research methodology corresponds to the stated goals.

The results confirm the importance of the chosen theme, demonstrating a different aggregation of the pathology depending on the location of the lesion and age, with the identification of prolonged postoperative status for the tumor locations in the cardiac area.

Author Response

Thank you for your kind comments. 

We will submit the revised manuscript, check it please.

Thank you very much. 
